# A synergetic perspective on urban scaling, urban regulatory focus and their interrelations

## Hermann Haken[1] and Juval Portugali[2]

[1]Institute for Theoretical Physics, Center of Synergetics, Stuttgart University, Stuttgart, Germany
[2]Department of Geography and the Human Environment, The Raymond and Beverly Sackler Faculty of Exact Sciences, School of Geosciences, Tel Aviv University, Tel Aviv, Israel

JP, 0000-0002-9526-6121

psychology/systems theory/complexity

**Keywords:**
urban scaling, regulatory focus theory, synergetic inter-representation networks, information adaptation

**Author for correspondence:**
Juval Portugali
e-mail: juval@tauex.tau.ac.il

By means of rich data, studies on urban scaling suggested that many urban properties scale with city size in universal ways. A recent study suggested an explanation why the behaviour of citizens in small and large cities differs qualitatively, by deriving the urban agents' behaviour from an extended version of Higgins' *regulatory focus theory* regarding humans' motivational system. Based on several sets of laboratory experiments, this study demonstrated that urban context of large, fast-paced cities and that of small slow-paced cities encourage two distinctively different motivations and behaviours on the part of their inhabitants. What remains an open question following the above study, however, is the way these behavioural reactions are related to the dynamics of cities as complex, adaptive, self-organization systems. The aim of the present paper is to answer this open question. It does so from the theoretical perspective of *Synergetics* and its application to the domain of cities by means of *synergetic inter-representation networks*, *information adaptation* and their conjunction. From this conjunction, the paper suggests a theoretical interpretation associated with a mathematical model that links the theoretical framework to the empirical findings.

## 1. Introduction

### 1.1. Urban allometry

The notion of urban allometry suggests that, similar to natural-organic complex systems, in cities too it can be shown statistically 'that important demographic, socioeconomic, and behavioural urban indicators are, on average, scaling functions of city size that are quantitatively consistent across different nations and times' [1].

Based on data from the USA, Germany and China, Bettencourt *et al.* [1] have proposed, firstly, that most urban indicators can be determined in terms of the following scaling law:

$$Y(t) = Y_0(t)N(t)^{\beta},$$

where $Y(t)$ stands for a given urban indicator, $N(t)$ is the population size of a city at time $t$ and $Y_0(t)$ is a time-dependent normalization constant. Secondly, that the scaling exponent $\beta$ that characterizes the various urban indicators, can take three universal forms: $\beta < 1$, a *sublinear regime* that typifies economies of scale associated with infrastructure and services (e.g. road surface area); $\beta \approx 1$, a *linear regime* associated with individual human needs (e.g. housing or household electrical consumption); and $\beta > 1$, a *superlinear regime* associated with outcomes from social interactions (e.g. income, number of patents etc.).

The title of Bettencourt *et al.*'s paper [1], 'Growth, innovation, scaling, and the pace of life in cities', indicates that cities differ in their 'pace of life', namely, that there are fast- and slow-paced cities—a view that goes back to the classical urban theories of Simmel, Wirth, Milgram and others [2–5]. Bettencourt *et al.*'s allometric approach attempted to quantify this view, relate it to city size, and show statistically that (contrary to organic systems) 'the pace of urban life is predicted to increase with [city] size'. As they demonstrated, this property shows itself in the various urban indicators that scale superlinearly ($\beta > 1$) with city size, ranging from innovation and wealth creation to crime rates, rates of spread of infectious diseases such as AIDS, and even pedestrian walking speed.

In a subsequent study, Bettencourt [6] rederived the scaling laws in a direct way. Based on a set of four assumptions regarding resources accessible to citizens, infrastructure, human mentality and social interaction, he predicted 'scaling exponents for a wide variety of urban indicators, from patterns of human behaviour and properties of infrastructure to the price of land' [6, p. 1439]. By this, he suggested, his modelling approach 'ties together the most microscopic needs and behaviours of individuals anywhere to the most macroscopic aspects of the urban infrastructure' [6, (S19)].

When scrutinizing Bettencourt's recent studies, we arrived at the conclusion that they rest on (at least) two preconditions: cities are self-similar fractal structures and they are subject to laws of large numbers; the implication: the approach applies to large enough cities, but not to small city/town sizes.

An alternative interpretation of the relations between the macroscopic properties of urban scaling and the microscopic aspects of urban agents' behaviour was recently suggested by Ross & Portugali [7]. Their paper explains 'why' and 'how' the size of cities has an effect on their inhabitants' and users' behaviour. Commencing bottom-up from humans' basic cognitive capabilities, they answer the 'why' and 'how' by extending the principles of Higgins' [8] *regulatory focus theory* (RFT) regarding humans' motivational system, to the context of cities. RFT and its extension to cities as *urban regulatory focus* (URF) are described next.

## 1.2. Regulatory focus theory

Higgins' RFT regarding humans' motivational system demonstrates that individuals' goal-directed behaviour is regulated by two distinct, and independently operating, motivational systems—*promotion* and *prevention* [8]. It further shows that while every person is driven by both promotion and prevention, promotion-oriented individuals are assertive, focus on winning and tend to take risks in order to achieve their goals, whereas prevention-oriented individuals are non-assertive, tend to avoid risks and to focus on not losing.

Higgins' RFT with its promotion–prevention tendencies refers to individuals' basic personal character and it is thus also termed *chronic regulatory focus*. Subsequent studies demonstrated that a person's chronic regulatory focus is context dependent, namely, that groups have an effect on the regulatory focus strategies of their members. In particular, Faddegon *et al.* [9] demonstrated empirically that the likelihood of one's behaving in a promotion or prevention way depends not only on one's personal (chronic) regulatory focus, but also on the atmosphere of the group one belongs to. That is, an assertive high-tech company (e.g. Apple company with its 'think different' slogan) and a conservative insurance company (whose possible slogan might be 'better safe than sorry') affect differently the personal regulatory focus of their employees. Based on these findings, Faddegon *et al.* [9] have suggested that promotion and prevention can characterize whole groups, thus giving rise to a *collective regulatory focus*. This effect was found in the context of small groups where face-to-face interaction prevails as well as in the case of larger groups where face-to-face interaction is rare.

## 1.3. Urban regulatory focus

Inspired by the above research, Ross & Portugali [7] theorized that regulatory focus processes occur at the urban level and suggested a link between RFT and the statistical findings of urban allometry: as noted

above, according to urban allometry studies [1,6], large cities are *usually* characterized by fast-paced and competitive urban dynamics, compared to small cities which are *usually* relaxed and slow-paced with a focus on safety, stability and security.

To test the above theorization, Ross & Portugali [7] have conducted a set of laboratory experiments in which each subject was, firstly, tested for his/her chronic regulatory focus (by means of Higgins *et al.*'s [10] questionnaire); secondly, was shown an (image of) urban scene of a fast-paced or a slow-paced city; and thirdly, was once again tested for his/her regulatory focus. This second test was implemented by means of the *recognition memory task* (a variation of a 'signal detection task'), which together with the *response bias* measure has been successfully used in the past in the context of RFT studies [9,11]. In the present set of experiments, the dependent variable was the participants' *response bias* that measured *the extent to which fast-paced cities* versus *slow-paced cities affected the personal (chronic) regulatory focus of the subjects.* In the model we develop below (§4), we denote the response bias as $b$. As noted above, while every person is driven by both promotion and prevention, some people are promotion oriented while others prevention oriented. Higher values of the response bias measure (e.g. higher $b$) indicate an increase in the personal promotion-oriented component due to the urban effect, while lower values (lower $b$), a decrease of the promotion component and an increase of the prevention component due to the urban effect.

Using the above methodological apparatus in their series of laboratory experiments, Ross & Portugali [7] have demonstrated that the promotion urban context of fast-paced cities (which are *usually* large cities) tend to intensify both promotion- and prevention-focused behaviours, thus motivating individuals to behave in extreme and polarized ways, whereas slow-paced cities (which are *usually* the small-sized urban environments), tend to encourage relatively more moderate and less polarized behaviour.

Note that we use and emphasize the expression 'usually' in the above paragraphs. This is to indicate that there are exceptions—small but fast-paced cities (e.g. Oxford, Silicon valley, …). Urban scaling studies, such as Bettencout *et al.*'s, refer to generic cases and general statistical regularities, to which there are always exceptions. Ross & Portugali [7], and thus our present study, refer essentially to slow versus fast-paced cities irrespective of their size, when the link to city size is due to [1,6].

## 1.4. Aims

The situation so far is as follows: from urban scaling studies we learn that urban allometry with its linear, superlinear and sublinear statistical regularities is a basic characteristic of cities as complex adaptive systems [1]. From Bettencourt's [6] subsequent study, as well as from other studies [12,13], we further learn that the above statistical regularities can be explicitly derived out of the property that cities as complex systems are self-similar fractal structures. This rederivation, however, is implicitly pre-conditioned by the laws of large numbers with the implication that the solution excludes the very small cities and towns in the urban system. Finally, from Ross & Portugali [7] study about URF, we learn that large and small cities affect the prevention–promotion tendencies of their inhabitants and users in different ways as above. What still remains an open question following these studies, however, is the way (or the extent to which) these motivational–behavioural reactions are related to the dynamics of cities as complex, adaptive, self-organization systems. How the dynamic of cities of different sizes is affected by, and is affecting, the promotion and prevention tendencies of their inhabitants and users?

Our aim in this paper is to 'close the circle' and answer this open question. We do so from the theoretical perspective of *synergetics*—Haken's [14] theory of complex self-organization systems—and its application to the domain of complexity theories of cities (CTC) by means of the notions of synergetic inter-representation networks (SIRN), information adaptation (IA) and their conjunction (SIRNIA). According to these theoretical perspectives, and as has been demonstrated in some details [15,16], urban dynamics is characterized by an ongoing interaction between external information/data conveyed by the urban environment and internal information that originates in urban agents' mind/ brain—a process captured by the notion of SIRN; in this process each urban agent adapts the incoming information/data from the city, by means of information previously constructed in the agent's mind/brain, as well as by the agent's chronic regulatory focus (i.e. its cognitive motivational inclination)—a process captured by the notion of IA.

Applied to the context of the present study, the external data/information is the size and properties of cities of various sizes as found by urban allometry studies, while the internal information is the promotion–prevention tendencies of the urban agents. The process of SIRNIA here refers to the way urban agents adapt to the information conveyed by the various urban environments, the effects of this adaptation process on their motivation, action and decision-making in the city and the resultant changes that take place in the city.

Our discussion below starts (in §2) with a short reminder of synergetics, SIRN, IA and their conjunction, first in general and then in the context of the dynamics of a city. Section 3 that follows suggests an SIRNIA perspective on individual URF and on its role in the dynamics of cities. In §4, we introduce a mathematical model constructed, on the one hand, on the basis of the theory developed in §3, while on the other, on the empirical findings of Ross & Portugali [7]. The paper concludes with suggestions for further research.

# 2. Synergetics, SIRNIA and the city—a reminder

## 2.1. Synergetics

This interdisciplinary field of research studies how the interaction between many individuals (participants, components, etc.) can produce specific macroscopic structures obeying their own laws [14]. This means that these structures act in specific ways to determine the behaviour of the individuals who (or which) contribute by their behaviour to that of the whole system—i.e. circular causality. As has been shown by synergetics, the whole process can be dealt with by means of two basic concepts and their mathematical formalization: (1) *order parameter(s)* (OP), (2) *slaving principle*. An OP (or very few of them) is an observable, in many cases measurable, quantity that *describes* the state of the total system. Simultaneously, the OP *prescribes* the behaviour of the individuals ('slaving principle'—a terminus technicus that has nothing to do with slaving!). A third important concept is the *control parameter* (CP). Similar to the OP, it is an observable, in many cases measurable, quantity, which, when it grows beyond a certain threshold, has the potential to destabilize the OP and thus leads to a phase transition and to the emergence of a new OP. An efficient way to exemplify the dynamics of synergetics is by means of the *laser paradigm* that is described graphically in figure 1.

Explicit mathematical applications of the laser paradigm to urban context were suggested by Weidlich [18] who employed the original mathematical formalism of synergetics in studies on public opinion and inter- and intra-urban migrations. Portugali [15,19] studied socio-cultural spatial segregation in cities by means of cellular automata and agent-based urban simulation models, employing synergetics and its laser paradigm as the interpretive theory. He has further suggested the latter interpretive theory in the study of specific urban phenomena such as the New York lofts and Tel-Aviv closed balconies [16].

Some concrete examples may help us to illuminate the range of applications: local concentrations of professions such as tailors in one street; restaurants at a specific square; artists living in the same quarter. Wealthy people build their houses in the same quarter, poor people in others. Racial discrimination leads to segregation. In all these cases, the relevant OP is the number of people having the same attitude or particular feature that distinguishes them from all other people. In this context, the slaving principle means that it is advantageous (in a material or even immaterial context (reputation)) to follow the main stream of the 'group' one belongs (or feels) to. In some cases, we may interpret the slaving principle as a social pressure. The control parameters are quantifiable 'features' such as land prices, rental rates, to mention but few. Quite remarkably, based on the laser paradigm, equations for the order parameters can be (and have been) established (see above). This kind of approach is continued and developed further in §4 below.

Beyond the above studies, examples of urban analogies are plentiful and exist on many time and length scales, ranging from spontaneously arising, self-regulated human behaviour in the form of applause after a concert when the irregular clapping of hands gives way to a synchronized clapping; or the *la-ola* wave in a stadium; through traffic on a city highway with speed limit, where at low vehicle density one observes uncorrelated motion, at medium density, uniform motion, while at elevated density, density waves and traffic jam; to cases of Matthew effects where the rich become richer (see [20] in this connection); to the phenomenon of *latent demand* [21] referring, for example, to the urban reality in which traffic congestion cannot be solved by introducing new roads, because putting in a new road just brings to the front latent demand that is activated by the new road link. (We are thankful to an anonymous reviewer for suggesting this example.)

## 2.2. Synergetic inter-representation networks and information adaptation

Cities are *dually complex* in that the city as a whole and each of its parts—the urban agents—is a complex system [15,16]. Looked upon from the perspective of the above laser paradigm, we have two self-organization

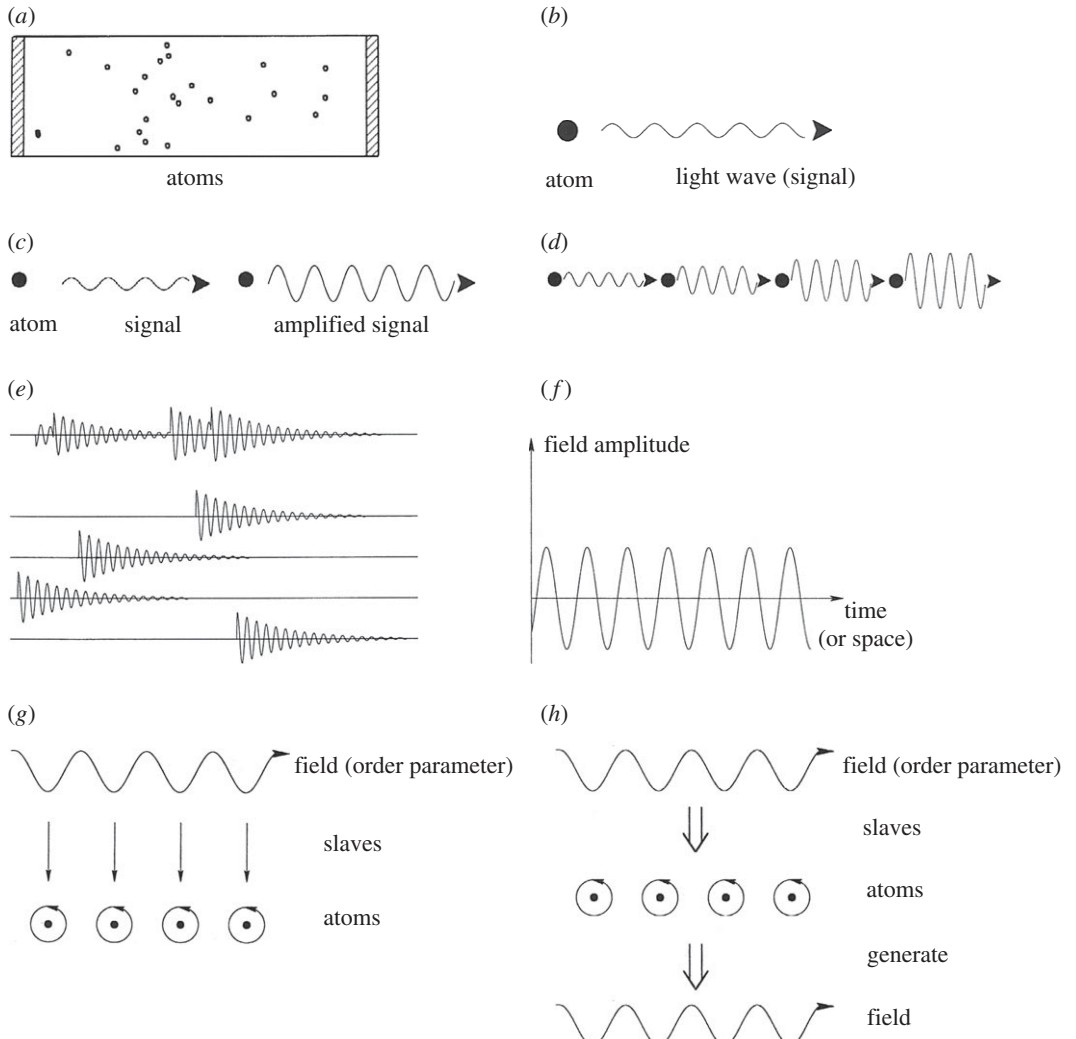

**Figure 1.** The laser paradigm. (*a*) Typical set-up of a gas laser. A glass tube is filled with gas atoms and two mirrors are mounted at its end faces. The gas atoms are excited by an electric discharge. Through one of the semi-reflecting mirrors, the laser light is emitted. (*b*) An excited atom emits light wave (signal). (*c*) When the light wave hits an excited atom, it may cause the atom to amplify the original light wave. (*d*) A cascade of amplifying processes. (*e*) The incoherent superposition of amplified light waves produces still rather irregular light emission (as in a conventional lamp). When sufficiently many waves are amplified, they strongly compete for further energetic supply. That wave that amplifies fastest wins the competition initiating laser action. (*f*) In the laser, the field amplitude is represented by a sinusoidal wave with practically stable amplitude and only small phase fluctuations. The result: a highly ordered, i.e. coherent, light wave is generated. (*g*) Illustration of the slaving principle. The field acts as an order parameter and prescribes the motion of the electrons in the atoms. The motion of the electrons is thus 'enslaved' by the field. (*h*) Illustration of circular causality. On the one hand, the field acting as order parameter enslaves the atoms. On the other hand, the atoms by their stimulated emission generate the field [17].

processes: one at the scale of the city as a whole, in which the agents, by means of their interaction, give rise to the city in a bottom-up manner, and the city acting as OP enslaves the behaviour of its agents in a top-down manner; and so on in circular causality. The other self-organization process takes place in each individual agent, but constrained by rules, laws, etc. of the city. The SIRN process (figure 2*a*) captures this process.

Cities are also *hybrid* complex systems in that they are composed of artefacts (buildings, roads, neighbourhoods, cities, etc.) which are simple systems, and human agents, each of which, as just noted, is itself a complex system. Urban agents thus interact not only among themselves but also with the urban artefacts. The process of IA (figure 2*b*) employs information theory in order to capture the information exchange between urban artefacts and the human agents. The notions of SIRN and IA were introduced in some details in the past [15,16,22,23]; here, we focus on their conjunction (SIRNIA) as it is associated with urban dynamics.

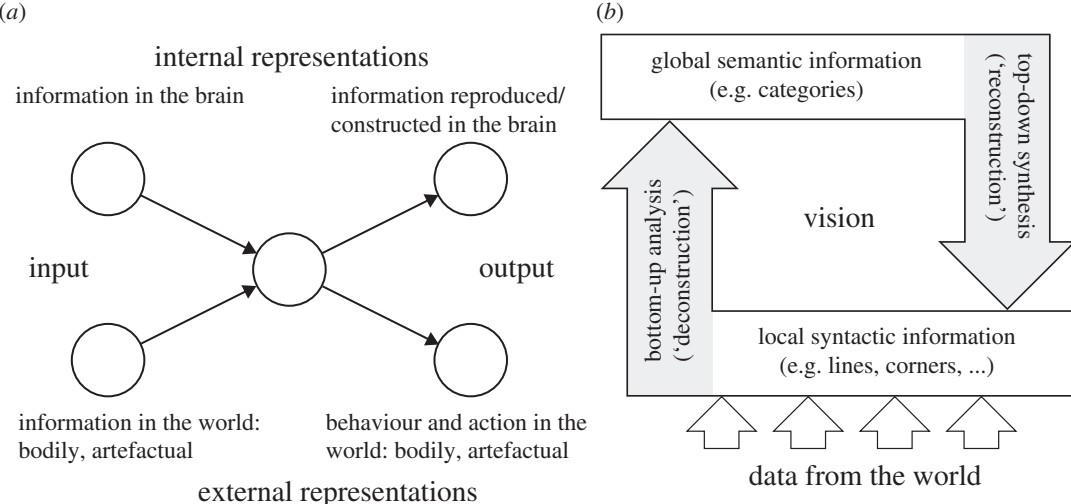

**Figure 2.** (*a*) The basic SIRN model symbolizes a self-organizing agent that is subject to two forms of information: external information that comes from the world (in our case, from the city) and internal information that 'comes' from the agent's mind/brain. The interaction between these two flows of information gives rise to two forms of information, again internal and external: internal information that is constructed by and in the mind/brain and external information in the form of, e.g. action in the city. (*b*) The basic process of *information adaptation* (IA) complements SIRN by elaborating on the way the mind–brain–body (MBB) handles the data that come from the world (e.g. from the city): such data are first analysed by the brain, in a bottom-up manner giving rise to local quantitative SHI syntactic information; this local information triggers a top-down process of synthesis that gives rise to qualitative (semantic and/or pragmatic) global information—in the case of vision, for example, to seeing and recognition. In this process, if the data that come from the city is too little, the mind/brain adds data (information inflation), if too much/superfluous, it extracts/ignores data (information deflation). See [22,23] for details.

In the context of urban dynamics, the notion of SIRNIA refers to the interaction between urban agents and their city. It suggests that an urban agent is ongoingly subject to two flows (figure 3): a flow of *data* that comes from the city and a flow of *information* that originates in, and comes from, the agent's mind/brain/memory. By means of the latter, and in a bottom-up manner, the agent's brain first transforms the incoming data flow into a quantitative, 'syntactic', Shannonian information (SHI); this syntactic SHI triggers a top-down process that transforms the quantitative SHI into a qualitative semantic or pragmatic forms of information (SI and PI, respectively). SI refers to the meaning *per se* (i.e. 'this is a chair'), whereas PI to the action possibilities afforded by an entity, in a way similar to Gibson's [24] notion of *affordances* (i.e. 'this object affords seating'). For a detailed discussion and bibliography on the relations between SHI, semantic information and pragmatic information, see [22,23].

## 3. A SIRNIA view on URF

The notion of SIRNIA, as noted, suggests that in the interaction between urban agents and their city, they are subject to two flows: a bottom-up flow of data that comes from the city and a top-down flow of information that originates in, and comes from, the agent's mind/brain/memory. We suggest that this process applies also to the experiments conducted by Ross & Portugali [7]. Here, the bottom-up flow of data came not from the city itself, but from images of cities that were screened to the subjects. In line with our basic SIRNIA model (figure 3), the brain first transformed the data into syntactic SHI, which triggered a top-down process that originated (1) in the person's previous knowledge, memory and experiences and (2) in the subject's chronic regulatory focus. Based on (1), the subject pattern recognized (and categorized) the screened city as fast-paced or slow-paced (thus transforming the SHI into SI) and then, based on an interaction between the pattern-recognized image and (2), the subject's mind/brain adapted its chronic regulatory focus (that can be seen as a potential PI) to the pattern-recognized fast-paced or slow-paced city, thus producing the materialized PI. In a real urban dynamics, this will take the form of action and behaviour in the city, whereas in the Ross & Portugali [7] laboratory experiment, this took the form of the subject's *response bias*. The latter measures the extent to which fast-paced and slow-paced cities affect the chronic promotion or prevention tendencies of the subjects.

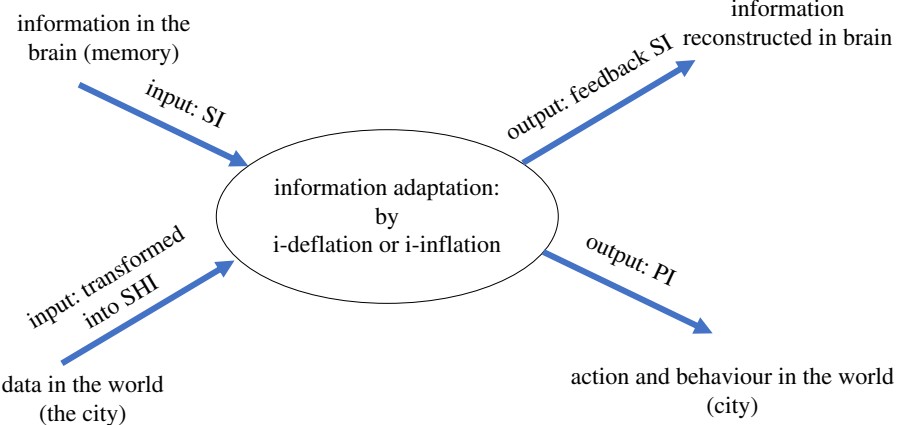

**Figure 3.** The basic SIRNIA model. For details, see text.

According to Ross & Portugali's [7] empirical findings, these PI adaptation processes, measured by means of the subjects' response bias, were as follows: when the city was pattern recognized as fast-paced, promotion-oriented subjects tended to exhibit a more promotion-focused motivation/behaviour (higher response bias), whereas prevention-oriented individuals tended to exhibit a more prevention-focused motivation and behaviour slower response (lower response bias). On the other hand, when the city was pattern recognized as slow-paced, those effects were eliminated. 'Participants with a dominant promotion focus displayed responses that were equally conservative as displayed by those with a dominant prevention focus' [7, p. 9].

According to synergetics and by implication to SIRNIA, by means of their interaction with each other, the city's inhabitants bottom-up give rise to the emergence of a city's OP, that once it comes into being, describes and prescribes (enslaves) the behaviour of the citizens in a top-down manner and so on in circular causality. Within this context, Ross & Portugali's [7] empirical findings refer to the second, top-down aspect of the process—demonstrating that the city top-down affects the regulatory focus of its inhabitants and users. The question is how? We'll answer this question by developing a new urban model that explicitly theorizes about and integrates RFT to the dynamic of cities.

## 4. Outline of the model

From Ross & Portugali [7] study follows that urban context of large, fast-paced cities encourages promotion-focused behaviour; adding to this finding, the logic of synergetics' circular causality process, it can be said that the encouraged promotion focus behaviour intensifies the fast-paced dynamics of the city, which in turn reinforces and further strengthens a pattern of promotion-focused behaviour, and so on and on in circular causality. This circular process may explain why in large, vibrant cities people take more risk, do more business, but also crimes, earn more money, produce and spend more.

As noted above (§1.3), in their laboratory experiments, Ross & Portugali [7] have used the *response bias* as a measure of the extent to which fast-paced cities versus slow-paced cities affected the personal (chronic) regulatory focus of the subjects. We denote the response bias variable by '$b$', where $b \geq 0$. Since we are dealing with an ensemble of citizens (represented by the test persons), we have to deal with the (relative frequency) distribution function

$$P(b;t), \tag{4.1}$$

where

$$\int_0^\infty P(b;t)\mathrm{d}b = 1.$$

In their paper, Rose & Portugali [7] measure the mean $M$

$$\bar{b} = \int_0^\infty P(b;t)\mathrm{d}b. \tag{4.2}$$

And the standard deviation $\sigma$ (denoted in [7] by SD); $\sigma$ is defined as square root of the variance

$$S = \overline{b^2} - \bar{b}^2, \tag{4.3}$$

i.e. $\sigma = S^{1/2}$. Since our approach becomes much simpler, in terms of $S$, we will work with $S$.

How can we calculate $P$ (4.1) or an equation for it in spite of the fact that we do not have any information on the processes on the 'microscopic' level, e.g. social contacts between people etc. but only the 'sparse data' (4.2), (4.3)? To solve this problem, we resort to the method of making unbiased guesses on complex systems when only few data (sparse data) are known. Starting from Jaynes' maximum (information) entropy principle and assuming a Markov process, Haken [17] derived a Fokker–Planck–Îto equation of the distribution function $P(b;t)$. A rederivation of this equation is beyond the scope of our paper. All we need is the explicit form of the drift and diffusion coefficients (or functions) based on the results of our paper; we present them in the appendix, cf. (A 1).

Quite remarkably, we need not present equation (A 1) explicitly in the main text, because we may derive from appendix (A 1) equations for the mean $\bar{b}$ (4.2) and the variance $S$ (4.3), cf. (A 5) and (A 6), and perform our analysis on the level of these equations. After having provided the reader with this 'background information,' we start from the equations

$$\frac{\mathrm{d}\bar{b}}{\mathrm{d}t} = -\gamma\bar{b} + h(t) \tag{4.4}$$

and

$$\frac{\mathrm{d}S}{\mathrm{d}t} = -2\gamma S + 2Q(t), \tag{4.5}$$

where, for the time being, the right-hand sides are considered as hypotheses. To verify them and in particular to derive explicit expressions for $h(t)$ and $Q(t)$, we proceed in two steps (which, in a way, resemble a 'gedanken' experiment). In the first step, we assume that we deal with people whose behaviour is not influenced by the city as a whole, but determined by their character and their reactions to typical random events. We assume that their behaviour can be modelled by a Fokker–Planck equation for which $h(t) = \gamma b_0$, and $Q(t) = Q$, where $\bar{b}$ is an average bias, whereas $\gamma$ is the inverse time after which an individual returns to his/her 'normal' state after some random incident. In the now following second step, we deal with the interplay between citizens and the city context so that we arrive at explicit expressions for $h(t)$ and $Q$. Here we must take care of the circular process described, in particular, at the beginning of this section. We consider the following two processes.

(1) The urban context $U$ determines the behaviour $b$, i.e. $P(b;t)$.

$$U \rightarrow P. \tag{4.6}$$

As specified above in §3 and figure 3, in terms of SIRNIA, the 'input' to this process is two information flows: the agent's 'chronic regulatory focus' and the information coming from the urban context; the 'output' is the agent's response bias which emerges in the process of IA, that is, the interaction between the two input information flows.

(2) The behaviour of the citizens determines the urban context $U$, i.e.

$$P \rightarrow U. \tag{4.7}$$

In contrast to the process (4.6) referring to an individual, here we consider the *collective* output of all citizens giving rise to urban context and, interestingly, to a collective mental state ('attitude' in our case).

In the following, we will make equations (4.6) and (4.7) explicit based on nonlinear Fokker–Planck–Îto equation (cf. appendix). (The nonlinearity enters via the dependence of $U$ on $P$, see below).

## 4.1. Impact of $U$ on $P$

This impact can be taken care of by means of equations (4.4) and (4.5). $U_M$ acts as additional 'force' $F(t)$ on $\bar{b}$ in (4.4), so that we identify

$$F(t) = U_M, \tag{4.8}$$

where, because of (4.7), $U_M$ is a function or functional of $P(b;t)$. This dependence will be discussed below, §4.2. Since $P$ is fixed by $\bar{b}$, and $S$, $U_M$ can only depend on these quantities. According to §4.2, we may approximate $U_M$ by

$$U_M \approx a\bar{b}^2. \tag{4.9}$$

There is also an impact of $U$ on $P$ via equation (4.5), or written as a formula

$$\frac{\mathrm{d}S}{\mathrm{d}t} = -2\gamma S + 2Q + U_S, \tag{4.10}$$

where, again, because of equation (4.7), $U_S$ is function(al) of $P$, which we approximate by

$$U_s = c\bar{b}, \tag{4.11}$$

where $c$ is a constant. Thus, equation (4.10) becomes

$$\frac{\mathrm{d}S}{\mathrm{d}t} = -2\gamma S + 2Q + c\bar{b}. \tag{4.12}$$

We will solve equation (4.3) with equations (4.8) and (4.12) in §4.3.

## 4.2. Impact of $P$ on $U$

As mentioned above, $U_M$ and $U_s$ must be functional of $P(b;t)$ which can be expressed by $\bar{b}$ and $S$. In this way, $U_M$ and $U_s$ become functionals of $\bar{b}$ and $S$ at several times, i.e. of $b(t)$, $b(t')$ etc. (In fact, an individual adapts more quickly than a whole city).

We now make the assumption that the citizens adapt quickly so that we may neglect time delays such as $t$–$t'$, etc. In other words, $U_M$ and $U_s$ become simple functions of $\bar{b}(t)$ $S(t)$, e.g.

$$U_M = U_M(\bar{b}(t),\, S(t)). \tag{4.13}$$

Under the reasonable assumption that the impact of $\bar{b}(t)$, $S(t)$, is not too large, we may approximate $U_M$ and $U_s$ by low-order polynomials of $\bar{b}$, $S$, where the impact of the size of $\bar{b}$ is far more dominant than that of $S$: an 'observer' (new citizen) is far more impressed with the strength of activities (e.g. speed) rather than with their variety. Consequently,

$$U_M \approx a_0\bar{b} + a\bar{b}^2, \tag{4.14}$$

When we insert equation (4.14) in equations (4.4) and (4.8), we observe that the term $a_0\bar{b}$ of equation (4.14) can be combined with the term $-\gamma\bar{b}$ in equation (4.4) so that, de facto, nothing has changed. Thus, we may drop $a_0\bar{b}$ or just include it in $-\gamma\bar{b}$. All in all we arrive at our first fundamental model equation

$$\frac{\mathrm{d}\bar{b}}{\mathrm{d}t} = -\gamma(\bar{b} - b_0) + a\bar{b}^2. \tag{4.15}$$

We turn to $U_s$ which is a measure of the restlessness, fluctuations, noise—in our case of a city/village. Here again, the size of $\bar{b}$ is the dominant cause, rather than the variation $S$ of the distribution of $b$. Thus we arrive at our second fundamental model equation

$$\frac{\mathrm{d}S}{\mathrm{d}t} = -2\gamma S + 2Q + c\bar{b}, \tag{4.16}$$

where it suffices to approximate $U_s$ by a linear term $c\bar{b}$.

## 4.3. Solution to the fundamental equations

Equations (4.15) and (4.16) can be solved in two steps. First we solve equation (4.15) exactly. Then we insert the result in equation (4.16), which can be solved exactly (though perhaps (?) not in closed form). For our present purpose, it suffices to treat the steady state solution to the equations (4.15) and (4.16), i.e.

$$-\gamma(\bar{b} - b_0) + a\bar{b}^2 = 0 \tag{4.17}$$

and

$$-2\gamma S + 2Q + c\bar{b} = 0. \tag{4.18}$$

The quadratic equation (4.17) possesses two solutions (bifurcation!):

$$\bar{b}_1 = \frac{\gamma'}{2} + \frac{\gamma'}{2}\left(1 - \left(\frac{4}{\gamma'}\right)b_0\right)^{1/2} \tag{4.19}$$

and

$$\bar{b}_2 = \frac{\gamma'}{2} - \frac{\gamma'}{2}\left(1 - \left(\frac{4}{\gamma'}\right)b_0\right)^{1/2}, \tag{4.20}$$

where

$$\gamma' = \frac{\gamma}{a}.$$

Concomitant with these solutions are $S_1$, $S_2$. By inserting $\bar{b}_1, \bar{b}_2$ into equation (4.18), we obtain

$$S_1 = \left(\frac{1}{2\gamma}\right)(2Q + c\bar{b}_1). \tag{4.21}$$

And, similarly

$$S_2 = \left(\frac{1}{2\gamma}\right)(2Q + c\bar{b}_2). \tag{4.22}$$

## 4.4. Making contact with observed data

Our model contains five parameters, $\gamma$, $b_0$, $a$, $Q$, $c$. As it transpires from the form of equations (4.15) and (4.16), $\gamma$ plays the role of a relaxation constant, or in other words, $\gamma$ is the inverse of an adaptation time, $\tau$. Since this is much shorter than the lifespan of an individual or even a city, we may assume that $\gamma$ is large. Furthermore, we may measure the constants $a$, $C$, $Q$ in terms of $\gamma$. This entails that we may formally put

$$\gamma = 1, \tag{4.23}$$

(this means that we measure time in units $1/\gamma$).

So we have to relate $b_0$, $a$, $Q$, $c$ to observed data. We denote them by $\bar{b}_1(0)$, $S_1(0)$, $\bar{b}_2(0)$, $S_2(0)$. Using equations (4.19–4.23), we readily obtain

$$\frac{1}{a} = \bar{b}_2(0) + \bar{b}_1(0) > 0, \tag{4.24}$$

$$a\bar{b}_1(0)\bar{b}_2(0) = b_0, \tag{4.25}$$

$$c = 2(\bar{b}_1(0) - \bar{b}_2(0))^{-1}(S_1(0) - S_2(0)) \tag{4.26}$$

and

$$Q = (S_1(0) + S_2(0)) - (S_1(0) - S_2(0))(\bar{b}_2(0) + \bar{b}_1(0))(\bar{b}_1(0) - \bar{b}_2(0))^2. \tag{4.27}$$

It remains to discuss how to obtain the data of the mean and standard deviation $\sigma$ (or equivalently $S = \sigma^2$) of the response bias.

(a) By means of field experiments with populations of large/small cities.
(b) As studies in Ross & Portugali [7] paper by 'laboratory' experiments.

Here we use the latter approach and analyse its data in the light of our model.

We observe that the relations (4.24–4.27) allow us to calculate the model parameters $a$, $b$, $c$, $Q$ from measured data, while, conversely, the relations (4.19–4.22) allow us to 'predict' the observed data on the basis of the model parameters. To illustrate our further procedure, we analyse 'the effect of urban context on promotion-focused participants' [7]. We denote this case by 1a) promotion low.

**Table 1.** Calculated parameters $b_0$, $a$, $c$, $Q$.

| Case | $b_0$ | $a$ | $c$ | $Q$ | $\bar{b}_1$ | $\bar{b}_2$ | $S_1$ | $S_2$ |
|------|-------|-----|-----|------|-----------|-----------|-------|-------|
| 1a | 0.064 | 4 | 0 | 0.01 | 0.15 | 0.11 | 0.012 | 0.012 |
| 1b | 0.084 | 3 | 0.06 | −0.004 | 0.15 | 0.18 | 0.014 | 0.017 |
| 2a | 0.087 | 3 | 0.8 | −0.052 | 0.16 | 0.18 | 0.012 | 0.020 |
| 2b | 0.064 | 4 | 0.2 | −0.001 | 0.15 | 0.11 | 0.014 | 0.010 |
| 3a | 0.09 | 2.8 | 0.3 | −0.01 | 0.16 | 0.2 | 0.014 | 0.02 |
| 3b | 0.051 | 4.6 | 1.2 | −0.042 | 0.14 | 0.08 | 0.12 | 0.008 |

According to [7]

Small city:  mean $M = 0.15$,  standard deviation $\sigma = 0.11$
Big city:    $M = 0.11$,      $\sigma = 0.11$

We denote the mean of small city by $\bar{b}_1$ and of big city by $\bar{b}_2$. We use the variance $S = \sigma^2$ with indices 1, 2 corresponding to $\bar{b}_1, \bar{b}_2$. Thus

$$\bar{b}_1 = 0.15, \ \bar{b}_2 = 0.11, \ S_1 = 0.012, \ S_2 = 0.012$$

Using equations (2.24)–(2.27), we obtain

$$b_0 = 0.064, a = 4, c = 0, \ Q = 0.01$$

We apply this procedure to all cases reported by Ross & Portugali [7]:

1a) promotion focus low, 1b) high
2a) prevention focus low, 2b) high
3a) dominant focus: promotion, 3b) prevention.

Our results are listed in table 1.

While the results on $M$, $\sigma$ (or in our notation $\bar{b}$, $S$) were discussed by Ross & Portugali, scrutinizing table 1 leads us to some new insights based on the parameters $b_0$, $a$, $c$. First of all, $c > 0$ means that larger bias $b$ leads to higher fluctuations in accordance with our above hypothesis. The result that in all cases $\bar{b}_1 > b_0$, $\bar{b}_2 > b_0$ means that living in a community enhances the personal bias; however, differently depending on his/her regulatory focus. Concerning $b_0$ and $a$, we note a remarkable symmetry between 1a) promotion focus low, and 2b) prevention focus high. This means, that low promotion focus and high prevention focus have the same effect on $\bar{b}_1, \bar{b}_2$. On the other hand, the fluctuations of $S_1$ are smaller in case 1a) than in 2b).

Approximately the same symmetry holds in the cases 1b), 2a), but this time including the fluctuations. In the case 3a), $b_0$ is still somewhat larger than that of case 1b) as can be expected. We may interpret $a b_0$ as 'effective' social interaction 'constant'. And, in fact, in all cases 1a) till 3b), this product yields practically the same value! While our model captures qualitatively and quantitively how the city context changes the mean bias of persons and the variance/standard deviation, we are aware that it was based on experimental data of 201 graduate students from the University of Tel Aviv. Surely, more data and studies are thus needed to validate the model and to draw more general conclusions about the behaviour of persons living in large and small cities.

## 4.5. Stability of solutions

To bring out the essentials, we consider equation (4.15) for the special case $b_0 = 0$, i.e.

$$\frac{d\bar{b}}{dt} = -\gamma\bar{b} + a\bar{b}^2. \tag{4.28}$$

The steady state solutions are

$$\bar{b}_2 = \frac{\gamma}{a} \tag{4.29}$$

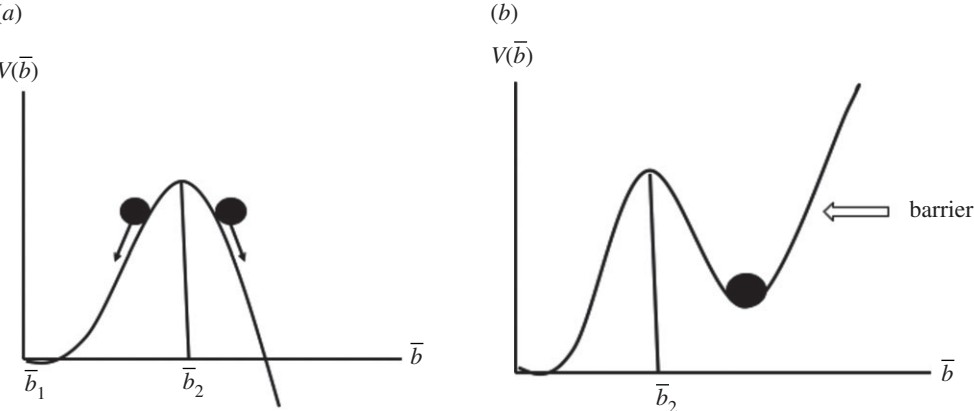

**Figure 4.** (*a*) Potential 'landscape'. (*b*) Potential 'landscape' with a barrier.

and

$$\bar{b}_1 = 0. \tag{4.30}$$

To check their linear stability, we insert $\bar{b}_2 = (\gamma/a) + \varepsilon, \quad \bar{b}_1 = \eta$ in equation (4.30) and obtain (in linear approximation)

$$\frac{d\varepsilon}{dt} = \gamma\varepsilon \tag{4.31}$$

and

$$\frac{d\eta}{dt} = -\gamma\eta, \tag{4.32}$$

while $\bar{b}_1$ represents a stable solutions, $\bar{b}_2$ is unstable. What does this imply? To this end, we rewrite equation (4.24) by means of a potential $V$ in the form

$$\frac{d\bar{b}}{dt} = -\frac{\partial V}{\partial \bar{b}}, \quad V = \gamma\frac{\bar{b}^2}{2} - \frac{a}{3}\bar{b}^3. \tag{4.33}$$

Figure 4 shows a plot of $V$. In terms of figure 4, the position of the ball on the slope of a hill symbolizes the size of $\bar{b}$ and its further development. Evidently the ball on the right slope of figure 4*a* indicates that $\bar{b}$ will increase indefinitely, i.e. that the average response bias increases forever—surely in contrast to human nature. Thus, in order to apply our model to a real situation, we must invoke a barrier such as drawn in figure 4*b*. Mathematically, the barrier can be taken care of by an additional term in equation (4.33), whose explicit form is not important for our discussion. As can be seen, $\bar{b}_2$ is a lower limit for people who want to belong to the 'large city' group.

Note, firstly, that in the light of these considerations and the results of Ross & Portugali's [7] paper, a high enough promotion focus seems to be a necessary prerequisite for the formation of the 'large city' group. Secondly, that the just-mentioned behaviour of $\bar{b}$, which can be read from figure 4 *left* and *right*, is fully substantiated by the exact time-dependent solutions to equation (4.15) (including $b_0 \neq 0$).

## 4.6. Summary of the model

In our above model, we have dealt with the following phenomenon that results, as noted above, from Ross & Portugali study: urban context of large, fast-paced cities encourages promotion-focused behaviour; adding to this empirical finding the logic of synergetics' circular causality process, it can be said that the encouraged promotion focus behaviour intensifies the fast-paced dynamics of the city, which in turn reinforces and further strengthens a pattern of promotion-focused behaviour, and so on. Thus, we are confronted with a problem of circular causality in a system with many participants, a problem at the core of Haken's synergetics [6]. In the parlance of synergetics [14], $\bar{b}$ plays the role of an OP and equation (4.15) is the OP equation. To tackle this specific problem, following a suggestion made in Ross & Portugali [7], we have chosen the 'response bias' of people, $b$, as the characteristic variable. Our central task has been to determine the equation for the probability distribution function

$P(b;t)$ that develops in the course of time, $t$, because of the circular process, and solve this equation. To this end, we have performed the following steps:

1. Since we aim at a comparison with measured data (cf. [7]) that are the mean value $M$ and standard deviation $\sigma$ and are compared to the complete distribution function $P(b;t)$ *sparse data*, we invoked a method that allows us to make the best guess on an equation for $P(b;t)$ under the conditions of given $\bar{b}$ and $\sigma$ or $S$ [17] (for a short summary cf. the appendix). This leads us to a Fokker–Planck–Îto equation.
2. Instead of solving this partial differential equation, we derive from it equations for $\bar{b}$ and $S$, first for people not in urban context.
3. We extend the $\bar{b}$ and $S$ equation by taking the urban context into account.

At each step, we make well-defined and well-justified approximations so that our final model equations rest on safe ground. In accordance with observed data [7], we obtain two distinct solutions corresponding to the large and small city cases. In particular, we have found in accordance with Ross & Portugali [7], that a high chronic promotion focus is a precondition for the formation of a 'large city' group.

## 5. Conclusion

Our aim in this paper was to study the way humans' basic motivational–behavioural tendencies are related to the dynamics of cities as complex, adaptive, self-organization systems—the ways the dynamics of cities of different sizes is affected by, and is affecting, the promotion and prevention tendencies of their inhabitants and users. We explored these issues from the theoretical perspective of synergetics approach to complexity theories of cities through the notions of SIRN, IA and their conjunction (SIRNIA). From these theoretical perspectives, we firstly suggested a descriptive account and then a mathematical model; both illustrate the circularly causal process by which urban contexts of large, fast-paced and slow-paced cities affect and are affected by the promotion- and prevention-focused behaviour of their citizens. As specified above, the urban process associated with this circularly causal process applies to both very large and very small cities and towns and includes the various ingredients of cities as complex systems: self-organization, a bottom-up emergence and qualitative phase-transition changes. Note that our approach sheds light on the creativity of evolutionary processes, in which people's error-making, mutations and innovations are also a result of evolution. While interesting and significant, these issues should await a subsequent study.

Data accessibility. No original data were used in this paper. The paper is composed of a theoretical discussion followed by a mathematical model. Both refer to the following empirical study that was recently published in RSOS: http://dx.doi.org/10.1098/rsos.171478 [7].

Authors' contributions. The two authors designed and performed the research, wrote the paper and have contributed equally to the conceptual framework of this paper. H.H. developed the mathematical model.

Competing interests. We have no competing interests.

Funding. No funding supported this research.

## Appendix A. Basic equations

*Wanted*: an equation for the best fit of a probability distribution function $P(b;t)$, of which only mean and standard deviation are known.

   *Answer:* making an 'unbiased guess under constraints' allows us to derive a Fokker–Planck–Îto equation for $P(b;t)$, which in the present special case reduces to the Fokker–Planck equation

$$\frac{\mathrm{d}P(b;t)}{\mathrm{d}t} = -\frac{\mathrm{d}}{\mathrm{d}b}\left((-\gamma b + h(t))P(b;t)) + Q(t)\frac{\mathrm{d}^2(P(b;t))}{\mathrm{d}b^2}\right), \tag{A 1}$$

where $h$ and $Q$ are independent of $b$.

   In the absence of urban context, we may put $h(t) = \gamma b_0$, $Q(t) = \mathrm{const}$.

   As we derived in the main text, in the case of urban context, we may put

$$h(t) = \gamma b_0 + U_M, \quad Q(t) = Q + \frac{1}{2}U_S, \tag{A 2}$$

where most importantly, $U_M$, $U_S$ are constants independent of $b$ (but depending on $\bar{b}$). Thus, $U_M$, $U_S$ do

not change the structure of (A 1). This equation allows us to derive equations for the mean

$$\bar{b} = \int_0^\infty b P(b;t)\mathrm{d}b, \tag{A 3}$$

and variance

$$S = \int_0^\infty b^2 P(b;t)\mathrm{d}b - \bar{b}^2, \tag{A 4}$$

from which we may calculate the standard deviation $\sigma$ by $\sigma = S^{1/2}$. To this end, we multiply (A 1), by $b$ or $b^2$, respectively, and integrate both sides over $b$. After partial integrations, we readily obtain

$$\frac{\mathrm{d}\bar{b}}{\mathrm{d}t} = -\gamma\bar{b} + h(t) \tag{A 5}$$

and

$$\frac{\mathrm{d}S}{\mathrm{d}t} = 2\gamma S + 2Q(t). \tag{A 6}$$

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
