## [Reviewer comments · Royal Society Open Science]

Review History

RSOS-181960.R0 (Original submission)

Review form: Reviewer 1 (Peter Allen)

Is the manuscript scientifically sound in its present form?

Yes

Are the interpretations and conclusions justified by the results?

Yes

Is the language acceptable?

Yes

Is it clear how to access all supporting data?

Yes

Do you have any ethical concerns with this paper?

No

Have you any concerns about statistical analyses in this paper?

No

Recommendation?

Accept as is

Comments to the Author(s)

This is an interesting and important paper, because it proposes a basis for a realistic link between the micro and macro behaviour that underlies self-organization within cities. This is usually posed in abstract terms where for example random differences at the local level lead to instabilities that in turn lead to the emergence of structure and macroscopic characteristics of urban systems. The key idea is that the urban context affects people's behaviour, and that people's behaviour leads to the urban context. This is an important insight and to a testable outcome where people with positive Higgins focus tend to live in big, bustling cities, while those with more caution tend to live in smaller, less dynamic urban centres. This is confirmed by the evidence cited in the paper.

Beyond this paper, I feel that these ideas will also help in understanding the pattern of the occurrence and diffusion of innovations, which are also involved in shaping the urban hierarchy. It seems likely therefore that Higgins' regulatory focus theory, may play a role in explaining the dynamic theory and pattern of innovation. Some cities seem to continue for very long periods to expand and change over time, while others, such as Detroit for example, can evolve through successful growth to painful decline, as real innovations cease to pervade them. So, I think that this micro-macro link put forward in this paper is important and should be published.

Review form: Reviewer 2

Is the manuscript scientifically sound in its present form?

No

Are the interpretations and conclusions justified by the results?

No

Is the language acceptable?

Yes

Is it clear how to access all supporting data?

Not Applicable

Do you have any ethical concerns with this paper?

No

Have you any concerns about statistical analyses in this paper?

No

Recommendation?

Major revision is needed (please make suggestions in comments)

Comments to the Author(s)

Attached file (Appendix A).

Review form: Reviewer 3

Is the manuscript scientifically sound in its present form?

No

Are the interpretations and conclusions justified by the results?

No

Is the language acceptable?

Yes

Is it clear how to access all supporting data?

Not Applicable

Do you have any ethical concerns with this paper?

No

Have you any concerns about statistical analyses in this paper?

No

Recommendation?

Reject

Comments to the Author(s)

Review:

A synergetic perspective on Urban scaling, Urban Regulatory Focus and their interrelations

The paper proposes a basic hypothesis on the nature of large-fast cities to small-slow cities, and then proceeds to relate scaling to this observation.

On page 4, the authors say:

Large cities are usually characterized by fast-paced competitive urban dynamics, compared to small cities which are usually relaxed and slow-paced with a focus on safety, stability, and security.

This seems too simplistic and contradictory to agglomeration economics observations that while some small cities (say with diverse economic functions, but no primary leading economic function) could indeed be relaxed, slow-paced and focus on safety, stability, and security, some others (say with a very specific economic function, such as highly automated manufacturing etc.) may be extremely fast-paced and volatile: the city is subject to large and fast fluctuations because it is tethered to the global economy, and global variations would have a large effect on the functioning of the city. So, not all large cities are the same, and not all small cities are the same.

2.1 Synergetics

The demonstration of the LASER - please add a paragraph showing an urban analogy. For example, if there is enough demand, a new public transit line is created. But because the new public transit line is created, it creates more residential demand around it, etc. In transport terminology, the idea is called latent demand - the tendency that congestion can't be solved by introducing new roads, because putting in a new road just brings to the front latent demand that is activated by the new road link.

2.2 SIRNIA

The idea of SIRNIA needs to be more clearly expressed. As it is now written, it is not clear exactly what is meant. For example: Is "I can use a chair as a door stopper" SI or PI? And why? The mapping appears very subjective and arbitrary. Maybe clearer explanation needed.

Later author's describe non-affordable objects, but what about the transformations of functions and affordances? In the case of a chair as a door stopper, the presumably the SI remains the same, but the PI has changed. How is the change in information expressed?

Critique of strong AI approaches: can be extremely brittle, since the number of possibilities can never be captured exhaustively.

In the urban context, what does "fast" or "slow" pace mean? Is it:

1. the walking speed of individuals
2. travel times
3. the speed of formation or dissolution of new firms
4. the rates of in or outmigration of goods or people or financial flows
5. the rates of economic processes, such as manufacturing
6. any number of other variables measuring "fast" or "slow"

My concern here is the following: If I have a city that is very small, but is very fast in the sense that it is a specialist manufacturing city for example, is it fast or slow? Or consider that travel times increase as a city gets larger, and if congestion sets in, it will take longer in a larger city to cover the same distance than in an uncongested smaller city - so does that mean that larger cities are slower, and smaller cities faster, since below a critical threshold, congestion will never appear?

I found that the entire assumption behind the model is highly simplistic, and does not relate to urban reality, since there are just way too many exceptions to the rules that are stated here. If we take these rules as given, then the math for the model follows, so I will not write much about that. But, the theoretical basis is too simplistic.

Decision letter (RSOS-181960.R0)

01-Apr-2019

Dear Professor Portugali:

Manuscript ID RSOS-181960 entitled "A synergetic perspective on Urban scaling, Urban Regulatory Focus and their interrelations" which you submitted to Royal Society Open Science, has been reviewed. The comments from reviewers are included at the bottom of this letter.

In view of the criticisms of the reviewers, the manuscript has been rejected in its current form. However, a new manuscript may be submitted which takes into consideration these comments.

Please note that resubmitting your manuscript does not guarantee eventual acceptance, and that your resubmission will be subject to peer review before a decision is made.

Your resubmitted manuscript should be submitted by 29-Sep-2019. If you are unable to submit by this date please contact the Editorial Office.

on behalf of Professor Miles Padgett (Subject Editor)
openscience@royalsociety.org

Associate Editor Comments to Author:

Thank you for your submission. We have now received 3 reviewers' reports. Should you decide to resubmit the manuscript, please carefully address each of the comments and provide a point by point response.

Please note that it is the editorial policy of Royal Society Open Science to offer authors only one round of revision in which to address changes requested by referees. If the revisions are not considered satisfactory by the Editor or referees, the paper may be rejected, and not considered further for publication by the journal.

Reviewers' Comments to Author:

Reviewer: 1

Comments to the Author(s)

This is an interesting and important paper, because it proposes a basis for a realistic link between the micro and macro behaviour that underlies self-organization within cities. This is usually posed in abstract terms where for example random differences at the local level lead to instabilities that in turn lead to the emergence of structure and macroscopic characteristics of urban systems. The key idea is that the urban context affects people's behaviour, and that people's behaviour leads to the urban context. This is an important insight and to a testable outcome where people with positive Higgins focus tend to live in big, bustling cities, while those with more caution tend to live in smaller, less dynamic urban centres. This is confirmed by the evidence cited in the paper.

Beyond this paper, I feel that these ideas will also help in understanding the pattern of the occurrence and diffusion of innovations, which are also involved in shaping the urban hierarchy. It seems likely therefore that Higgins' regulatory focus theory, may play a role in explaining the dynamic theory and pattern of innovation. Some cities seem to continue for very long periods to expand and change over time, while others, such as Detroit for example, can evolve through successful growth to painful decline, as real innovations cease to pervade them. So, I think that this micro-macro link put forward in this paper is important and should be published.

Reviewer: 2

Comments to the Author(s)
Attached file.

Reviewer: 3

Comments to the Author(s)
Review:

A synergetic perspective on Urban scaling, Urban Regulatory Focus and their interrelations

The paper proposes a basic hypothesis on the nature of large-fast cities to small-slow cities, and then proceeds to relate scaling to this observation.

On page 4, the authors say:

Large cities are usually characterized by fast-paced competitive urban dynamics, compared to small cities which are usually relaxed and slow-paced with a focus on safety, stability, and security.

This seems too simplistic and contradictory to agglomeration economics observations that while some small cities (say with diverse economic functions, but no primary leading economic function) could indeed be relaxed, slow-paced and focus on safety, stability, and security, some others (say with a very specific economic function, such as highly automated manufacturing etc.) may be extremely fast-paced and volatile: the city is subject to large and fast fluctuations because it is tethered to the global economy, and global variations would have a large effect on the functioning of the city. So, not all large cities are the same, and not all small cities are the same.

2.1 Synergetics

The demonstration of the LASER - please add a paragraph showing an urban analogy. For example, if there is enough demand, a new public transit line is created. But because the new public transit line is created, it creates more residential demand around it, etc. In transport terminology, the idea is called latent demand - the tendency that congestion can't be solved by introducing new roads, because putting in a new road just brings to the front latent demand that is activated by the new road link.

2.2 SIRNIA

The idea of SIRNIA needs to be more clearly expressed. As it is now written, it is not clear exactly what is meant. For example: Is "I can use a chair as a door stopper" SI or PI? And why? The mapping appears very subjective and arbitrary. Maybe clearer explanation needed.

Later author's describe non-affordable objects, but what about the transformations of functions and affordances? In the case of a chair as a door stopper, the presumably the SI remains the same, but the PI has changed. How is the change in information expressed?

Critique of strong AI approaches: can be extremely brittle, since the number of possibilities can never be captured exhaustively.

In the urban context, what does "fast" or "slow" pace mean? Is it:

1. the walking speed of individuals

2. travel times
3. the speed of formation or dissolution of new firms
4. the rates of in or outmigration of goods or people or financial flows
5. the rates of economic processes, such as manufacturing
6. any number of other variables measuring "fast" or "slow"

My concern here is the following: If I have a city that is very small, but is very fast in the sense that it is a specialist manufacturing city for example, is it fast or slow? Or consider that travel times increase as a city gets larger, and if congestion sets in, it will take longer in a larger city to cover the same distance than in an uncongested smaller city - so does that mean that larger cities are slower, and smaller cities faster, since below a critical threshold, congestion will never appear?

I found that the entire assumption behind the model is highly simplistic, and does not relate to urban reality, since there are just way too many exceptions to the rules that are stated here. If we take these rules as given, then the math for the model follows, so I will not write much about that. But, the theoretical basis is too simplistic.

Author's Response to Decision Letter for (RSOS-181960.R0)

See Appendix B.

RSOS-191087.R0

Review form: Reviewer 1 (Peter Allen)

Is the manuscript scientifically sound in its present form?

Yes

Are the interpretations and conclusions justified by the results?

Yes

Is the language acceptable?

Yes

Do you have any ethical concerns with this paper?

No

Have you any concerns about statistical analyses in this paper?

No

Recommendation?

Accept as is

Comments to the Author(s)

I believe that my previous acceptance of the paper as it was perhaps did not take into account sufficiently the very interdisciplinary nature of the journal. Therefore, some additional paragraphs and clarifications that the authors have supplied were necessary. I do believe the topic is important because it considers the creative possibilities within an evolution, as the peoples decisions affect the outcomes and circumstances, and these both change as a result of the evolution. I have published some ideas in the past (late 1980s) concerning evolutionary 'drive', where the capacity of a population to evolve is affected and indeed organized by the evolution itself, and thus the pathways into the future are richer than most work is concerned with. In other words, the 'error-making, mutations, innovations, experiments of a population are also a result of evolution. So, the interplay of the success or failure of explorations/innovations etc. with the desire to explore or innovate is all part of the complexity of reality. This paper is one of the rare ones that actually moves beyond a mechanical, though non-linear, response of agents. For that reason it makes an important contribution.

Review form: Reviewer 2

Is the manuscript scientifically sound in its present form?

Yes

Are the interpretations and conclusions justified by the results?

No

Is the language acceptable?

Yes

Do you have any ethical concerns with this paper?

No

Have you any concerns about statistical analyses in this paper?

Yes

Recommendation?

Accept with minor revision (please list in comments)

Comments to the Author(s)

Dear Editor

The manuscript "A Synergetic Perspective on Urban scaling, Urban Regulatory Focus and their Interrelations", by H. Haken and J. Portugali, present a mathematical framework based on regulatory focus theory to study the behavior of citizens in small and large cities.

All the question were clarified by the author. But I am skeptical if the results are robust. So, I suggest a modification in the last paragraph of section 4.4, page 22, the authors wrote: "All in all our model captures qualitatively and quantitatively how the cities context changes the mean bias of persons and the variance/standard deviation."

The experimental data used in the present manuscript were based on 201 graduate students from the University of Tel-Aviv. I think it is too early to say that the empirical data are valid to draw conclusions about behavior of persons living in large and small cities. It is safer to state that more data and studies are needed to validate the model. However, there is some evidence based on the report of the experiment in the article [7].

Limitations of the mathematical model derivation, according to the answers of the last report, are related to the fact that available the data contain mean and standard deviation (article [7])

only. We do not know if the assumption will survive when confronted with large data sample. Finally, I suggest two references to credit others articles which derives models about statistical regularities of cities assuming self-similarity fractal structures (page 5, section 1.4) :

Chen, Y., & Zhou, Y. (2008). Scaling laws and indications of self-organized criticality in urban systems. *Chaos, Solitons & Fractals*, 35(1), 85-98.

Ribeiro, F. L., Meirelles, J., Ferreira, F. F., & Neto, C. R. (2017). A model of urban scaling laws based on distance dependent interactions. *Royal Society open science*, 4(3), 160926.

Minor correction: on page 18, formula 15 change $(b_0)^-$ for b_0 .

After this complementation, in my opinion, the manuscript is suitable to be publish on Royal Society Open Science journal.

Decision letter (RSOS-191087.R0)

16-Jul-2019

Dear Professor Portugali

On behalf of the Editor, I am pleased to inform you that your Manuscript RSOS-191087 entitled "A synergetic perspective on Urban scaling, Urban Regulatory Focus and their interrelations" has been accepted for publication in Royal Society Open Science subject to minor revision in accordance with the referee suggestions. Please find the referees' comments at the end of this email.

The reviewers and Subject Editor have recommended publication, but also suggest some minor revisions to your manuscript. Therefore, I invite you to respond to the comments and revise your manuscript.

- Ethics statement

- Data accessibility

If you wish to submit your supporting data or code to Dryad (<http://datadryad.org/>), or modify your current submission to dryad, please use the following link:
<http://datadryad.org/submit?journalID=RSOS&manu=RSOS-191087>

- Competing interests

- Authors' contributions

- Acknowledgements

- Funding statement

Because the schedule for publication is very tight, it is a condition of publication that you submit the revised version of your manuscript before 25-Jul-2019. Please note that the revision deadline will expire at 00.00am on this date. If you do not think you will be able to meet this date please let me know immediately.

- 1) A text file of the manuscript (tex, txt, rtf, docx or doc), references, tables (including captions) and figure captions. Do not upload a PDF as your "Main Document".
- 2) A separate electronic file of each figure (EPS or print-quality PDF preferred (either format should be produced directly from original creation package), or original software format)

- 3) Included a 100 word media summary of your paper when requested at submission. Please ensure you have entered correct contact details (email, institution and telephone) in your user account
- 4) Included the raw data to support the claims made in your paper. You can either include your data as electronic supplementary material or upload to a repository and include the relevant doi within your manuscript
- 5) All supplementary materials accompanying an accepted article will be treated as in their final form. Note that the Royal Society will neither edit nor typeset supplementary material and it will be hosted as provided. Please ensure that the supplementary material includes the paper details where possible (authors, article title, journal name).

on behalf of Prof Miles Padgett (Subject Editor)
openscience@royalsociety.org

Reviewer comments to Author:
Reviewer: 1

Comments to the Author(s)

I believe that my previous acceptance of the paper as it was perhaps did not take into account sufficiently the very interdisciplinary nature of the journal. Therefore, some additional paragraphs and clarifications that the authors have supplied were necessary. I do believe the topic is important because it considers the creative possibilities within an evolution, as the peoples decisions affect the outcomes and circumstances, and these both change as a result of the evolution. I have published some ideas in the past (late 1980s) concerning evolutionary 'drive', where the capacity of a population to evolve is affected and indeed organized by the evolution itself, and thus the pathways into the future are richer than most work is concerned with. In other words, the 'error-making, mutations, innovations, experiments of a population are also a result of evolution. So, the interplay of the success or failure of explorations/innovations etc. with the desire to explore or innovate is all part of the complexity of reality. This paper is one of the rare ones that actually moves beyond a mechanical, though non-linear, response of agents. For that reason it makes an important contribution.

Reviewer: 2

Comments to the Author(s)

Dear Editor

The manuscript "A Synergetic Perspective on Urban scaling, Urban Regulatory Focus and their Interrelations", by H. Haken and J. Portugali, present a mathematical framework based on regulatory focus theory to study the behavior of citizens in small and large cities.

All the question were clarified by the author. But I am skeptical if the results are robust. So, I suggest a modification in the last paragraph of section 4.4, page 22, the authors wrote: "All in all our model captures qualitatively and quantitatively how the cities context changes the mean bias of persons and the variance/standard deviation."

The experimental data used in the present manuscript were based on 201 graduate students from the University of Tel-Aviv. I think it is too early to say that the empirical data are valid to draw conclusions about behavior of persons living in large and small cities. It is safer to state that more data and studies are needed to validate the model. However, there is some evidence based on the report of the experiment in the article [7].

Limitations of the mathematical model derivation, according to the answers of the last report, are related to the fact that available the data contain mean and standard deviation (article [7]) only. We do not know if the assumption will survive when confronted with large data sample. Finally, I suggest two references to credit others articles which derives models about statistical regularities of cities assuming self-similarity fractal structures (page 5, section 1.4) :

Chen, Y., & Zhou, Y. (2008). Scaling laws and indications of self-organized criticality in urban systems. *Chaos, Solitons & Fractals*, 35(1), 85-98.

Ribeiro, F. L., Meirelles, J., Ferreira, F. F., & Neto, C. R. (2017). A model of urban scaling laws based on distance dependent interactions. *Royal Society open science*, 4(3), 160926.

Minor correction: on page 18, formula 15 change (b_0) for b_0 .

After this complementation, in my opinion, the manuscript is suitable to be publish on Royal Society Open Science journal.

Author's Response to Decision Letter for (RSOS-191087.R0)

See Appendix C.

Decision letter (RSOS-191087.R1)

23-Jul-2019

Dear Professor Portugali,

I am pleased to inform you that your manuscript entitled "A synergetic perspective on Urban scaling, Urban Regulatory Focus and their interrelations" is now accepted for publication in Royal Society Open Science.

You can expect to receive a proof of your article in the near future. Please contact the editorial office (openscience_proofs@royalsociety.org and openscience@royalsociety.org) to let us know if

you are likely to be away from e-mail contact. Due to rapid publication and an extremely tight schedule, if comments are not received, your paper may experience a delay in publication.

on behalf of Miles Padgett (Subject Editor)
openscience@royalsociety.org

Appendix A

Dear Editor

The authors Hermann Haken and Juval Portugali present the manuscript entitle "A synergetic perspective on Urban scaling, Urban Regulatory Focus and their interrelations." The goal of the present paper is to answer this open question: How the dynamic of cities of different sizes is affected by, and is affecting, the promotion and prevention tendencies of their inhabitants and users? To answer the question the paper suggests a mathematical model that link the theoretical framework to the empirical findings.

The manuscript is not clear. I had many doubts along the text. So, I have some questions: I will follow the section order.

1.2 Regulatory focus theory, second paragraph

"They demonstrated that the likelihood of one's behaving in a promotion or prevention way depends not only on one's personal regulatory focus, but also on the collective regulatory focus of the group one belongs to."

It is missing example of what is personal regulatory focus and collective regulatory focus which is related to the model on section 4. Please, give at least two examples related to small groups and large groups.

1.4 Aims

"What still remains an open question following these studies, however, is the way (or the extent to which) these motivational behavioral reactions are related to the dynamics of cities as complex, adaptive, selforganization systems. How the dynamic of cities of different sizes is affected by, and is affecting, the promotion and prevention tendencies of their inhabitants and users?"

What is the connection between this open question and the parameter b in the model? Please, give some example.

"Our aim in this paper is to 'close the circle' and answer this open question. We do so from the theoretical perspective of Synergetics ... urban dynamics is characterized by an on-going interaction between external information/data conveyed by the urban environment and internal information that originates in urban agents' mind/brain—a process captured by the notion of SIRN; in this process urban agents adapt the incoming information/data by previously constructed information and by their chronic cognitive motivational inclinations—a process captured by the notion of IA."

This paragraph is not clear.

a) what is SIRN and what is IA ? It is missing a clear definition here.

" in this process urban agents adapt the incoming information/data by previously constructed information and by their chronic cognitive motivational inclination. " This sentence is difficult to understand.

b) The agent adapt the information or adapt due the information or adapt to the information?

c) What and how is the previously constructed information?

d) What do you mean by "their chronic cognitive motivational inclination"?

2.1 Synergetics

Again, it is not clear what are:

- a) order parameter(s),
- b) slaving principle
- c) control parameter (CP).

The present definition is too general . The figure 1 does not help us to understand. It would be nice if the authors give example using cities. In section 4, they model the order parameter denoted by letter b (bias). They could anticipate an example of b here. A clear example, to help us to understand the modeling. Also, be explicit what is the slaving principle and the control parameter in the example.

2.2 SIRNIA

In caption of Figure 2 the authors say self-organizing agent . I know the concept of self-organized systems. All agent in cities are self-organizing agents?What is this? I think the authors should say just agent. What matter is that agents follow a set of rules to generate a self-organized systems.

Figure 2 left and Figure 3 are the same in my opinion. It is better to eliminate figure 2 or substitute Figure2 left by Figure 3.

3. A SIRNIA view on URF

“ when the city was pattern recognized as fast-paced, promotion oriented subjects tended to exhibit a more promotion focused motivation/behavior (higher response bias), whereas prevention-oriented individuals tended to exhibit a more prevention-focused motivation and behavior slower response (lower response bias). On the other hand, when the city was pattern recognized as slow-paced, those effects were eliminated. “Participants with a dominant promotion focus displayed responses that were equally conservative as displayed by those with a dominant prevention focus” [ibid,9]. According to synergetics and by implication to SIRNIA, by means of their interact”

What is the definition of higher/lower response bias?

Is promotion focused motivation the proxy for higher response bias?

Why more prevention-focused motivation was classified as behavior slower response? I think is better to supply more information to explain higher or lower response bias.

4. Outline of the model

On second paragraph we read: “As stated in [3], the adequate variable is the response bias that can be measured by psychological experiments and is defined by the following formula:”

a)What are p(false alarm) and p(hit)?

It is missing some information before introduce this equation for Br.

Rewrite the sentence: In their paper Rose and Portugali [3] measure the mean M and here the expression is denoted by \bar{b}

The same for the standard deviation: Here the author mention SD in paper ref3, starting with notation sigma and change to S. Why not go straight to the point and use S?

b) It is not clear the Fokker Plank derivation. It is missing a complete description of the process to derive the Fokker Planck equation. First question, what is the process which allowed the authors propose the equation A.1 ?

$$\frac{dP(b;t)}{dt} = -\frac{d}{db}((- \gamma b + h(t))P(b;t)) + Q(t)\frac{d^2(P(b;t))}{db^2}$$

The variables gamma, h(t) and Q(t) were not defined in the context of city and the process is unknown. Even if the author say that this is a general Fokker Planck equation, so, there is a more general expression for Fokker Planck, how to justify this choice? It is missing a fundamental description of the process in the context of cities.

c) So the authors say In the absence of urban context, we may put that $h(t) = \gamma b_0$,

$$h(t) = \gamma b_0 + U_M, Q(t) = Q + \frac{1}{2}U_S$$

Whats is absence of urban context? What is the Fokker Planck in the absence of urban context?

d) Where most importantly, U_M , U_S are constants independent of b (but depending on b_{bar}) . Why?

e) The author address the equation (4) and (5) to the appendix. In the some moment in the appendix the author address the section 4 to write h(t) as $h(t) = \gamma b_0$ or

$$h(t) = \gamma b_0 + U_M, Q(t) = Q + \frac{1}{2}U_S$$

The author say that:

“The urban context U determines the behavior b, i.e. $P(b;t)$.

$U \rightarrow P$

The behavior of the citizens determines the urban context U, i.e.

$P \rightarrow U$ ”

f) How to convert this information into equation? I expected a transcendental equation.

4.1 Impact of U on P

g) The authors say that U_M , U_S act as force F(t) on b_{bar} . What kind of interaction these forces describe? Why U_M depends only on $b_{bar}(t)$ and $S(t)$? It is not clear that it is enough neglect time delays. Please, give us more details.

h) According to Sect. 4.2, we may approximate U_M by...

I could not understand the argument on sect. 4.2. justify the approximation written as formula (9) and (11).

4.2 Impact of P on U

The third paragraph in this section is not clear. The authors say:

“An “observer” (new citizen) is far more impressed with the strength of activities (e.g. speed) rather than with their variety. Consequently”
I could not connect this argument with Formula (14).

What is the relation between strength of activity or variety with expression (14)?

4.3 Solution to the fundamental equations

In this section author study the steady state solution. I suppose that cities are not a system in equilibrium. How the authors guarantee the validity of such solution to cities dynamics?

4.4 Making contact with observed data

Here I expected the authors could use real data to validate the model. So, the question is, why I should belief in the present model? Why the author did not bring real data in this section? Is the present model a good model?

The manuscript is not clear and ready for publication. The point is, a lot of assumptions are made and no one can judge if they are correct or not. The authors should clarify the issues presented here and justify the model better. The ideal would be to identify a concrete case where variables and parameters were calibrated with real data.

Appendix B

Responses to the reviewers

We would like to thank the reviewers for the time spent in reviewing our manuscript, and for the comments and questions made, which helped us improve our manuscript. Please find below a point-by-point reply to all comments and questions. In what follows, the reviewers' quotations from our paper are in Black, their comments are in *italics*, while our responses in Red.

Response to Reviewer 1

We are delighted to read that reviewer 1 recommends to publish the paper as it is.

Responses to Reviewer 2

1.2 Regulatory focus theory, second paragraph

“They demonstrated that the likelihood of one’s behaving in a promotion or prevention way depends not only on one’s personal regulatory focus, but also on the collective regulatory focus of the group one belongs to.”

It is missing example of what is personal regulatory focus and collective regulatory focus which is related to the model on section 4. Please, give at least two examples related to small groups and large groups.

In p. 4 we’ve added explanations and examples.

1.4 Aims

“What still remains an open question following these studies, however, is the way (or the extent to which) these motivational behavioral reactions are related to the dynamics of cities as complex, adaptive, selforganization systems. How the dynamic of cities of different sizes is affected by, and is affecting, the promotion and prevention tendencies of their inhabitants and users?”

What is the connection between this open question and the parameter b in the model? Please, give some example.

The parameter b in the model is the *response bias* employed by Ross and Portugali in their laboratory experiments. Three new paragraphs about the response bias were added to Sect. 1.3 (pages 4-5) to explain the response bias and its connection and role in the present study.

“Our aim in this paper is to ‘close the circle’ and answer this open question. We do so from the theoretical perspective of Synergetics ... urban dynamics is characterized by an ongoing interaction between external information/data conveyed by the urban environment and internal information that originates in urban agents’ mind/brain—a process captured by the notion of SIRN; in this process urban agents adapt the incoming information/data by previously constructed information and by their chronic cognitive motivational inclinations—a process captured by the notion of IA.”

This paragraph is not clear.

a) *what is SIRN and what is IA ? It is missing a clear definition here.*

Two new paragraphs were added at the beginning of Sect. 2.2 to clarify the notions of SIRN, IA and their conjunction. See pages,9-10.

" in this process urban agents adapt the incoming information/data by previously constructed information and by their chronic cognitive motivational inclination. " *This sentence is difficult to understand.*

b) *The agent adapt the information or adapt due the information or adapt to the information?*

We have reformulated this sentence (p. 6) to clarify that each agent *adapts* the incoming data from the city. This process is further elaborated in Sect 2.2.

c) *What and how is the previously constructed information?*

We've reformulated the sentence (p.6) to clarify that we refer to memorized information previously constructed in an agent's mind/brain. See also Sect 2.2.

d) *What do you mean by "their chronic cognitive motivational inclination"?*

We've reformulated the sentence (p.6) to clarify that by "chronic cognitive motivational inclination" we refer to an agent's regulatory focus which is also explained in the newly added text to Sect. 1.2 p. 4.

2.1 Synergetics

Again, it is not clear what are:

a) *order parameter(s),*

b) *slaving principle*

c) *control parameter (CP).*

The present definition is too general . The figure 1 does not help us to understand.

Three new paragraphs were added to Sect. 2.1 (p.7-8): They further explain the meaning of a), b), c) above and provide concrete examples.

It would be nice if the authors give example using cities. In section 4, they model the order parameter denoted by letter b (bias). They could anticipate an example of b here. A clear example, to help us to understand the modeling. Also, be explicit what is the slaving principle and the control parameter in the example.

Two new paragraphs were added to Sect. 2.1 (pages 7-8) that give examples of explicit mathematical applications of the laser paradigm to urban context, as well as urban examples that have yet to be modeled. See also the additional sentence in the caption to Fig. 1.

2.2 SIRNIA

In caption of Figure 2 the authors say self-organizing agent . I know the concept of selforganized systems. All agent in cities are self-organizing agents? What is this? I think the authors should say just agent. What matter is that agents follow a set of rules to generate a self-organized systems.

As explained by the two new paragraphs added to Sect. 2.2 (p. 9-10), there is a difference between complex systems in the material, organic and city domains: In the first, the reviewer is right: the parts (agent) give rise to the self-organized system. In the second, each part (e.g. animal) is itself a complex self-organized system and this has to be taken into consideration. Applied to cities, it means that cities are dual complex systems: the city as a whole is a complex self-organized system and each urban agent is also a complex self-organized system. As further explained in the newly added paragraph (pages 9—10), cities further differ from both, as some of their parts (such as buildings, roads, etc.) are artifacts and as such simple systems with the implication that cities are hybrid complex systems.

Figure 2 left and Figure 3 are the same in my opinion. It is better to eliminate figure 2 or substitute Figure 2 left by Figure 3.

We don't think so as Fig. 2, left describes the SIRD process, (Fig. 2 right the IA process) while Fig 3 integrates SIRD+IA and thus describes the SIRDIA process.

3. A SIRDIA view on URF

“ when the city was pattern recognized as fast-paced, promotion oriented subjects tended to exhibit a more promotion focused motivation/behavior (higher response bias), whereas prevention-oriented individuals tended to exhibit a more prevention-focused motivation and behavior slower response (lower response bias). On the other hand, when the city was pattern recognized as slow-paced, those effects were eliminated. “Participants with a dominant promotion focus displayed responses that were equally conservative as displayed by those with a dominant prevention focus” [ibid,9]. According to synergetics and by implication to SIRDIA, by means of their interact”
What is the definition of higher/lower response bias?

As elaborated in the newly written Sect 1.3 (and at outset to Sect. 4), the *response bias* is a measure of the extent to which fast-paced cities versus small-paced cities affected the personal (chronic) regulatory focus of the subjects.

Is promotion focused motivation the proxy for higher response bias? Why more prevention-focused motivation was classified as behavior slower response? I think is better to supply more information to explain higher or lower response bias.

Following the newly added paragraphs in Sect. 1.3, this statement should now be clear.

4. Outline of the model

On second paragraph we read: “As stated in [3], the adequate variable is the response bias that can be measured by psychological experiments and is defined by the following formula:”

a) What are $p(\text{false alarm})$ and $p(\text{hit})$?

It is missing some information before introduce this equation for Br.

In light of the new paragraphs added to Sects. 1.3 and 4, about response bias, we believe that its formula at the outset of Sect. 4 is obsolete – we've thus deleted it.

Rewrite the sentence: In their paper Rose and Portugali [3] measure the mean M and here the expression is denoted by the same for the standard deviation: Here the author mention SD in paper ref3, starting with notation sigma and change to S . Why not go straight to the point and use S ?

We want(ed) to establish a link to the Ross and Portugali paper

b) It is not clear the Fokker Plank derivation. It is missing a complete description of the process to derive the Fokker Plank equation. First question, what is the process which allowed the authors propose the equation A.1 ?

The variables γ , $h(t)$ and $Q(t)$ were not defined in the context of city and the process is unknown. Even if the author say that this is a general Fokker Plank equation, so, there is a more general expression for Fokker Plank, how to justify this choice? It is missing a fundamental description of the process in the context of cities.

A historical note may be in order. Fokker (a PhD student of Planck) and Planck derived their equation in the context of the Langevin equation of Brownian motion, where the physical processes at the microscopic level are sufficiently well known. In our present “city” paper, the situation is quite different, because we don't have sufficient information on the (social, economic, ..) processes on the microscopic level. All we have is the number of people, the mean

value \bar{b} and the standard deviation. To derive, nevertheless, an equation, whose solution can be compared with experimental data, we resort to the method of making unbiased guesses on complex systems when only few data (“sparse data”) are known. Starting from Jaynes’ maximum (information) entropy principle, and assuming a Markov process, Haken (1988, 2000) derived a Fokker-Planck *Îto* equation of the distribution function $P(b;t)$ of the general form

$$\frac{dP}{dt} = -\frac{d}{db}(K(b;t)P) + Q(b,t)\frac{d^2P}{db^2} \quad (1)$$

If Q is independent of b , (1) reduces to a conventional Fokker-Planck equation.

Dear reviewer, for our further detailed explanation, please consult the newly formulated text to our paper, specifically, pages 13-19 and 19-21. We hope that we have sufficiently clearly answered your question b).

c) So the authors say In the absence of urban context, we may put that $h() = 0$, what is absence of urban context? What is the Fokker Planck in the absence of urban context?

The “bias” behavior of a person more or less shielded against the impact of social “pressures”. In this case, \bar{b} relaxes within time $1/\gamma$ to \bar{b}_0 .
The Fokker-Planck equation reduces to one with

$$K = -\gamma\bar{b} + \gamma b_0, \quad Q(t)=Q$$

d) Where most importantly, UM, US are constants independent of b (but depending on \bar{b}). Why?

We are working in the frame of the “sparse data” approach where we have knowledge only about \bar{b} !

e) The author address the equation (4) and (5) to the appendix. In the some moment in the appendix the author address the section 4 to write $h(t)$ as $h() = 0$ or The author say that:

“The urban context U determines the behavior b , i.e. $P(b;t)$.

$U \rightarrow P$

The behavior of the citizens determines the urban context U , i.e.

$P \rightarrow U$ ”

To clarify the above, we’ve added two new paragraphs to Sect. 4 and have altered eq (4). See pages 13-15.

f) How to convert this information into equation? I expected a transcendental equation.

4.1 Impact of U on P

This is explained in eqs. (7)—(16) of the model. Because we are dealing with mean-values, no transcendental equations are needed.

g) The authors say that UM, US act as force $F(t)$ on b_{bar} . What kind of interaction these forces describe? Why UM depends only on $b_{\text{bar}}(t)$ and $S(t)$? It is not clear that it is enough neglect time delays. Please, give us more details.

The “forces” represent the strength of social etc. interactions so to change \bar{b} . UM depends only on \bar{b} , $S(t)$ because these are the only measurable variables in the frame of the “sparse data” approach. The neglect of time-delays is justified because the adaptation time of citizens is much shorter than that of the whole city.

h) According to Sect. 4.2, we may approximate UM by...

I could not understand the argument on sect. 4.2. justify the approximation written as formula (9) and (11).

Our arguments have been provided in the text.

4.2 Impact of P on U

The third paragraph in this section is not clear. The authors say:

“An “observer” (new citizen) is far more impressed with the strength of activities (e.g. speed) rather than with their variety. Consequently”

I could not connect this argument with Formula (14).

What is the relation between strength of activity or variety with expression (14)?

We wanted to justify that U depends primarily on \bar{b} and not on S .

4.3 Solution to the fundamental equations

In this section author study the steady state solution. I suppose that cities are not a system in equilibrium. How the authors guarantee the validity of such solution to cities dynamics?

We apply the concept of time-scale separation: change of city dynamics is much slower than that of a citizen.

4.4 Making contact with observed data

Here I expected the authors could use real data to validate the model. So, the question is, why I should belief in the present model? Why the author did not bring real data in this section? Is the present model a good model?

The manuscript is not clear and ready for publication. The point is, a lot of assumptions are made and no one can judge if they are correct or not. The authors should clarify the issues presented here and justify the model better.

The ideal would be to identify a concrete case where variables and parameters were calibrated with real data.

Please note, firstly, that we have no data on “field experiments” in real cities. Rather, we use data gained by laboratory experiments with groups of people. Secondly, in our revised Sect. 4.4 we have calculated (“calibrated”) all parameters for all cases of “real” data and discussed the results showing that there is a good agreement between those data and our model with the implication that our model is a good one.

Responses to Reviewer 3

1

On page 4, the authors say:

Large cities are usually characterized by fast-paced competitive urban dynamics, compared to small cities which are usually relaxed and slow-paced with a focus on safety, stability, and security.

This seems too simplistic and contradictory to agglomeration economics observations that while some small cities (say with diverse economic functions, but no primary leading economic function) could indeed be relaxed, slow-paced and focus on safety, stability, and security, some others (say with a very specific economic function, such as highly automated manufacturing etc.) may be extremely fast-paced and volatile: the city is subject to large and fast fluctuations because it is tethered to the global economy, and global variations would have a large effect on the functioning of the city. So, not all large cities are the same, and not all small cities are the same.

In response to the above comments and in order to clarify the various issues, we've re-written Sects. 1.2 and 1.3 and added a new paragraph at the end of Sect. 1.3. See pages 3-5 in the attached.

2

2.1 Synergetics

The demonstration of the LASER - please add a paragraph showing an urban analogy. For example, if there is enough demand, a new public transit line is created. But because the new public transit line is created, it creates more residential demand around it, etc. In transport terminology, the idea is called latent demand - the tendency that congestion can't be solved by introducing new roads, because putting in a new road just brings to the front latent demand that is activated by the new road link.

Three new paragraphs were added to Sect. 2.1 showing, firstly, formal mathematical applications of the laser paradigm to urban phenomena; secondly, many urban analogies, including the 'latent demand' example suggested by the reviewer. A note was also added to the caption of Fig. 1. See pages 7—9 in the attached.

3

2.2 SIRNIA

The idea of SIRNIA needs to be more clearly expressed. As it is now written, it is not clear exactly what is meant. For example: Is "I can use a chair as a door stopper" SI or PI? And why? The mapping appears very subjective and arbitrary. Maybe clearer explanation needed.

Two new paragraphs were added at the beginning of Sect. 2.2 to clarify the notions of SIRN, IA and their conjunction. See pages 9-10.

Later author's describe non-affordable objects, but what about the transformations of functions and affordances? In the case of a chair as a door stopper, the presumably the SI remains the same, but the PI has changed. How is the change in information expressed?

Following this comment we realize that the side-remark (and its figure 3) about the non-affordable objects as PI does not clarify the issue and instead creates confusion. Since it is essentially anecdotal and not needed to the main discussion, we have decided to omit it.

4

Critique of strong AI approaches: *can be extremely brittle, since the number of possibilities can never be captured exhaustively.*

We didn't refer in our paper to AI or strong AI at all.

5

In the urban context,

what does "fast" or "slow" pace mean? Is it:

1. *the walking speed of individuals*
2. *travel times*
3. *the speed of formation or dissolution of new firms*
4. *the rates of in or outmigration of goods or people or financial flows*
5. *the rates of economic processes, such as manufacturing*
6. *any number of other variables measuring "fast" or "slow"*

A new paragraph was added to sect. 1.1 (pages 2-3) about pace of life in cities.

6

My concern here is the following: If I have a city that is very small, but is very fast in the sense that it is a specialist manufacturing city for example, is it fast or slow? Or consider that travel times increase as a city gets larger, and if congestion sets in, it will take longer in a larger city to cover the same distance than in an uncongested smaller city - so does that mean that larger cities are slower, and smaller cities faster, since below a critical threshold, congestion will never appear?

I found that the entire assumption behind the model is highly simplistic, and does not relate to urban reality, since there are just way too many exceptions to the rules that are stated here. If we take these rules as given, then the math for the model follows, so I will not write much about that. But, the theoretical basis is too simplistic.

As noted in the newly added paragraphs (pages 4-5), we are fully aware of exceptions—the existence of small but fast-paced cities (e.g. Oxford, Silicon valley, ...). Urban scaling studies such as Bettencout's et al, refer to generic cases and general statistical regularities, to which there are always exceptions. Ross and Portugali [7], and thus our present study and model, refer essentially to slow vs. fast paced cities irrespective of their size; the link to city size is due to studies [1, 6] that despite many exceptions, still indicate positive relations between city size and pace of life.

Appendix C

Responses to the reviewers

We would like to thank the reviewers for the time spent in reviewing our manuscript, and for the comments and questions made, which helped us improve our manuscript. Please find below a point-by-point reply to all final comments made by the reviewers toward publication.

Response to Reviewer 1

In response to Reviewer's 1 comments we've added the following sentence (with a footnote) at the end of the Conclusions (page 25): "Note that our approach sheds light on the creativity of evolutionary processes, in which people's error-making, mutations and innovations, are also a result of evolution.¹ While interesting and significant, these issues should await a subsequent study."

Responses to Reviewer 2

We've responded to all comments made by Reviewer 2:

As suggested by Reviewer 2, we've modified the last paragraph of Sect. 4.4 and it is now reads as follows: "While our model captures qualitatively and quantitatively how the city context changes the mean bias of persons and the variance/standard deviation, we are aware that it was based on experimental data of 201 graduate students from the University of Tel-Aviv. Surely, more data and studies are thus needed to validate the model and to draw more general conclusions about behavior of persons living in large and small cities.

In line with Reviewer 2 comment, we've added to Sect. 1.4 (page 5) the two references that derives models about statistical regularities of cities assuming self-similarity fractal structures.

Formula (15) was corrected as suggested.

¹ We thank an anonymous reviewer for this comment.